# After action review of the response to an outbreak of Lassa fever in Sierra Leone, 2019: Best practices and lessons learnt

**Charles Njuguna**[1]*, **Mohamed Vandi**[2], **Evans Liyosi**[1], **Jane Githuku**[1], **James Sylvester Squire**[2], **Ian Njeru**[1], **Ian Rufus**[1], **Victoria Katawera**[1], **Wilson Gachari**[1], **Robert Musoke**[1], **Claudette Amuzu**[1], **Mukeh Fahnbulleh**[2], **Joseph Bunting-Graden**[2], **Janet Kayita**[1], **James Bunn**[1], **Ambrose Talisuna**[3], **Zabulon Yoti**[3]

**1** World Health Organization Country office, Freetown, Sierra Leone, **2** Ministry of Health and Sanitation, Freetown, Sierra Leone, **3** World Health Organization Regional Office for Africa, Brazzaville, Congo

* njugunach@who.int

**Data Availability Statement:** All relevant data are within the manuscript and its Supporting Information files.

## Abstract

### Background

In November 2019, an outbreak of Lassa Fever occurred among health workers in a non-endemic district in Sierra Leone. The outbreak resulted in five cases, including two that were exported to the Netherlands. The outbreak tested multiple technical capacities in the International Health Regulations (2005) in a real-life setting. As such, an after action review (AAR) was undertaken as recommended by World Health Organization. We report on the findings of the AAR including best practices and lessons learnt.

### Methods

A two stage review process was employed. The first stage involved national pillar level reviews for each technical pillar and one review of the district level response. The second stage brought together all pillars, including participants from the national and sub-national level as well as health sector partners. National guidelines were used as references during the deliberations. A standardized template was used to report on the key findings on what happened, what was supposed to happen, what went well and lessons learnt.

### Results

This was a hospital associated outbreak that likely occurred due to a breach in infection prevention and control (IPC) practices resulting in three health workers being infected during a surgical operation. There was a delay in detecting the outbreak on time due to low index of suspicion among clinicians. Once detected, the outbreak response contained the outbreak within one incubation period. Areas that worked well included coordination, contact tracing, active case search and ring IPC. Notable gaps included delays in accessing local emergency funding and late distribution of IPC and laboratory supplies.

**Funding:** The World Health Organization Sierra Leone provided technical and financial support for the meetings that generated data for this article. WHO technical staff supported the Ministry of Health and Sanitation in study design, data collection, data analysis and manuscript preparation. Funding provided by WHO was from routine funds available at the country office and no special grant was received for this work.

**Competing interests:** The authors have declared that no competing interests exist.

## Conclusions

The incident management system worked optimally to contain this outbreak. The core technical gaps identified in surveillance, IPC and delay in deployment of resources should be addressed through systemic changes that can mitigate future outbreaks.

## Author summary

The International Health Regulations (IHR) Monitoring and Evaluation Framework was developed by the World Health Organization to provide strategies to monitor and assess how countries are building their core public health capacities under IHR (2005). The framework has four components: annual reporting on IHR capacities (mandatory), Joint External Evaluation, simulation exercises and After Action Review (AAR). In November 2019, an outbreak of Lassa Fever occurred among health workers in a non-endemic district in Sierra Leone. The outbreak resulted in five cases, including two deaths and two exported cases to the Netherlands. The outbreak tested multiple technical capacities in the IHR (2005) in a real-life setting. We therefore conducted an AAR to assess how well the country responded to the outbreak. This hospital associated outbreak likely occurred due to a breach in infection prevention and control (IPC) practices. The response launched after detection of the outbreak successfully contained the outbreak within one incubation period. Areas that worked well included coordination, contact tracing, active case search and ring IPC. Areas that needed improvement were clinicians' knowledge on Lassa Fever, delays in accessing local emergency funding and late distribution of IPC and laboratory supplies.

## Introduction

After Action Review (AAR) is a qualitative assessment of the actions taken in response to a public health event of concern [1] and is among the four components of WHO International Health Regulations (IHR) monitoring and evaluation framework [2–4]. The other components of the framework are simulation exercises, state party annual reports (SPAR) and joint external evaluations (JEE). AAR allows countries to assess the functionality of public health systems after an emergency and identify best practices that should be maintained and issues that need to be corrected. The overarching goal of the AAR process is to identify immediate, medium and long-term actions needed to increase IHR core capacities.

The Ebola outbreak in West Africa (2014–2016) triggered in depth reflections on the state of public health capacities in affected countries and issues that impaired an effective public health response [5–9]. Most reviews concurred on the need to improve sensitivity and timeliness of infectious disease surveillance systems, strengthen health systems in low and middle-income countries and test functionality of public health preparedness and response systems. The annual IHR state party self-assessments do not guarantee that the reported capability and functionality of the public health emergency response systems actually exists. Thus, the three most affected countries in the West African Ebola Outbreak displayed limitations in early detection of the outbreak and mounting effective responses, despite reporting fairly developed IHR capacities.

The need for more rigorous assessments of IHR core capacities and a shift to in-depth assessments of functionality led to the revision of the IHR Monitoring and Evaluation

Framework in 2016 [2] in line with the recommendations of the Second Extensions for Establishing National Public Health Capacities on IHR Implementation [10]. WHO developed guidelines and tools for structured AARs that if properly conducted can provide an opportunity for stakeholders who participated in a response to translate their experiences into lessons learnt. Since adopting the use of AAR in 2016, WHO has supported more than 16 after action reviews of public health events [4]. One such AAR was conducted in Sierra Leone after the mudslide emergency that led to the death of 1141 and displacement of approximately 5900 people in 2017 [11].

Following the 2014–2016 EVD outbreak, there has been considerable investment in the public health systems in Sierra Leone. Notably, the Integrated Disease Surveillance and Response (IDSR) system was revitalized starting in 2015 and has over time become well established in all public health facilities [12]. There is near 100% compliance by health facilities countrywide to submission of weekly disease surveillance data. In 2019, a shift to electronic health facility based surveillance reporting was completed and this reduced delays in reporting [13]. National and district level public health preparedness and response plans have been developed and structures operationalized. A national IPC program is now well established [14] and regular assessments of IPC status in health facilities is carried out. The functionality of the emergency preparedness and response structures post-Ebola outbreak has been tested severally such as during the mudslide emergency in 2017 [11] and through a full scale EVD simulation exercise in 2019 [15].

Two months following the end of the Lassa Fever outbreak in Tonkolili district in 2019, an AAR was undertaken, in line with WHO IHR Monitoring and Evaluation Framework and country requirements for learning from public health incidents. This paper describes the methods and the findings of the AAR which can be used to guide future AARs.

## The incident: Cluster of Lassa fever cases in a hospital in a non- endemic district

On 20th November 2019, the Sierra Leone Ministry of Health and Sanitation (MoHS) was notified by the Netherlands through the World Health Organization about a confirmed case of Lassa Fever, in a doctor who worked in a district in Northern Sierra Leone. The doctor had since been evacuated to the Netherlands on 19th November 2019 where he had tested positive for Lassa Fever and died 2 days later while receiving treatment. Upon receiving this notification, the MoHS dispatched national and district rapid response teams to verify the occurrence and extent of the outbreak.

Once the outbreak was verified, a rapid risk assessment was conducted and the incident was graded as level two, in line with the MOHS's National Incident and Emergency Response Plan (NIERP) and WHO's Emergency Response Framework (ERF) 2nd edition. The incident was graded as level two as it required coordination at the national level and multiple agencies were likely to participate in the response.

## Summary of Lassa fever outbreak investigation in tonkolili district

The outbreak investigations identified a cluster of five Lassa Fever cases: two probable cases and three confirmed cases. The two probable cases were patients admitted in Masanga Mission Hospital in Tonkolili district. The patients had undergone surgical operations that were carried out by the three health workers who were also confirmed as cases. Lassa Fever was not considered as a possible diagnosis, until it was detected in the Dutch doctor evacuated for treatment in Netherlands on 19 November 2019.

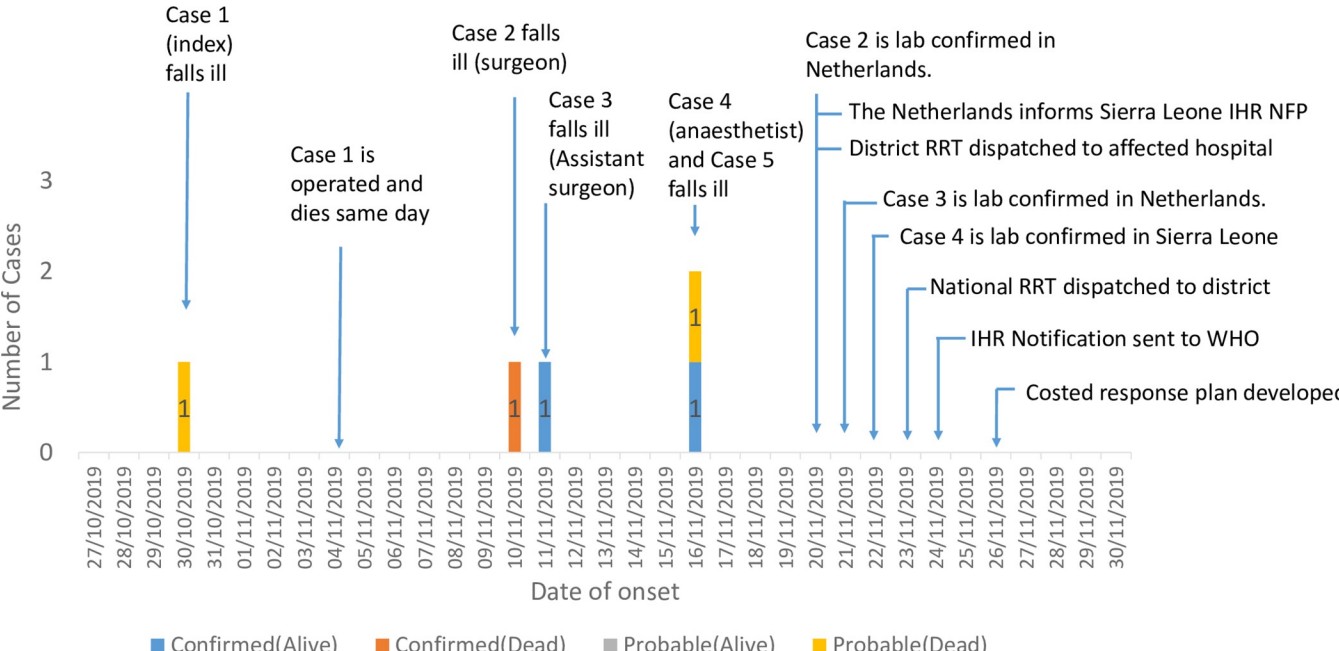

**Fig 1. Sequence of events for the Lassa fever outbreak, Tonkolili District, Sierra Leone, 2019.** IHR NFP-International Health Regulations National Focal Point; RRT- Rapid Response Team.

The likely primary (index) case was a pregnant woman referred to the mission hospital from a primary health care unit on 4[th] November 2019, with per vaginal bleeding and fever. She underwent a caesarian section on the same day. Three health workers, including the operating doctor, his assistant and the nurse anesthetist who participated in the surgical procedure, later developed signs and symptoms compatible with Lassa Fever (Fig 1). The fifth case occurred in a female patient who underwent an abdominal operation, in the same hospital theatre, by the same medical team on the same day as the primary case. The fifth case developed fever fourteen days after the surgical operation and died three days later. Having identified an epidemiologic link among all the cases, the investigating teams suspected nosocomial transmission, likely due to a breach in IPC protocols. Eighty-one persons who had been in contact with the cases were identified, quarantined and followed up. No secondary cases were identified among the contacts of the five cases.

Outbreak response activities included activation of the Incident Management System (IMS) at national and district level, cleaning and disinfection of the affected hospital (Masanga Hospital), refresher training on Infection Prevention and Control (IPC), distribution of IPC supplies, prompt case detection and isolation as well as quarantine of close contacts. The outbreak was successfully contained with a total of five (5) cases being identified.

## Methods

### Ethics statement

Authority to publish this work was obtained from the Ministry of Health and Sanitation, Sierra Leone. Ethical approval was deemed unnecessary from an ethical review board as it is not required for routine outbreak response activities undertaken by the Ministry of Health and Sanitation. Patient consent for publication was not required as no personal level data has been included.

## Study setting and design

This study was conducted in early 2020 in Sierra Leone following a response to a Lassa Fever outbreak that occurred in Tonkolili District, Northern Province, in late 2019. Its main goal was to assess how well the public health response was carried out by the national and district teams. After Action Review was therefore selected as the study design since it's one of the recommended methods by the World Health Organization for learning from incidents.

The AAR process was mainly guided by the country document *Learning from Incidents and Exercises*: *Guidance on EPRR Debriefing and After Action Reviews* (S1 File) which is a guide on how to conduct AARs that was adapted from the WHO guidelines for conducting AARs [1] The AAR review process utilized two processes; an internal organizational structured AAR and a multi-agency AAR. Both processes were conducted within the expected three months after the end of the Lassa Fever outbreak in Tonkolili district, as per the World Health Organization AAR guidelines [1].

The Lassa Fever outbreak response tested ten (10) of thirteen (13) IHR core capacities (Table 1). and therefore information relevant to these areas was collected during the AAR. The other 3 core capacities (Food safety, Chemical events, Radiation events) were not tested in this response and were therefore not a major focus of the AAR.

## Data collection and analysis

**AAR preparatory phase.** Preparations for the AAR began in January 2020, with formation of a review team made up of officials from MoHS, WHO and US-CDC. The review team identified an overall lead facilitator to guide the review process and consolidate the findings. The lead facilitator was an independent consultant hired by WHO country office and who had not taken part in the Lassa Fever response. This was in line with the country guidelines which requires that the lead facilitator be independent to ensure impartiality. The lead facilitator had many years' experience in conducting AARs and simulation exercises and was the consultant that helped the country to develop the guidelines that were used for conducting the AAR.

The incident management system for responding to the Lassa Fever outbreak was organized into six main pillars namely 1) Coordination; 2) Surveillance and laboratory; 3) Case management; 4) IPC, safe and dignified burial; 5) Risk communication and social mobilization; and 6) Logistics. The AAR also utilized the same pillar system so as to allow an exhaustive review of the strengths and gaps per pillar.

To help the lead facilitator in successfully conducting the AAR, national pillar level support facilitators were selected from MoHS and other Ministries, Departments and Agencies (MDAs) based on their technical expertise. As per the country guidelines, the support facilitators were selected from subject matter experts which helped to guide the discussions around themed areas that they were conversant with and this prevented deviation beyond the planned

**Table 1. IHR capacities tested by the Lassa fever outbreak, Tonkolili District, Sierra Leone, 2019.**

1. Legislation and Financing
2. IHR Coordination and National IHR Focal Point Functions
3. Zoonotic Events and the Human-Animal Interface
4. Laboratory
5. Surveillance
6. Human resources
7. Health service provision
8. National Health Emergency Framework
9. Risk Communication
10. Points of Entry

scope and objectives. To ensure impartiality and objectivity in the discussion, each pillar had team members drawn from various MDAs as well as partner organizations such as WHO, US CDC and other partners. To guide the review process, a presentation template was developed and shared for use during pillar level AAR. This template collected information on what was expected to be done, what was done, what went well, what did not go well and recommended areas for improvement (S2 File). Participants were also provided with copies of the country guidelines *Learning from Incidents and Exercises* and were trained on them so that they could familiarize themselves with the process of conducting the AAR.

### The AAR phase

We conducted the actual AAR in two phases. In the first phase of the review, six national pillar level and one district level after action reviews were held on 18[th] March 2020. Discussions were guided using six questions listed in Table 2.

When establishing how events unfolded, reference was made to technical reports on the outbreak response and minutes of PHNEOC coordination meetings. To gauge the appropriateness of the response, local guidelines, including the multi hazard response plans, PHNEOC Concept of Operations and the guidelines and operational procedures for rapid response teams in public health emergencies were referenced. Support facilitators collated all information from the discussions using the templates provided.

In the second phase of the review, a one-day multi-agency workshop that brought together stakeholders from MoHS, other MDAs, as well as key health sector partners was held on 20[th] March 2020. In this session, technical leads for each pillar and the District Medical Officer (DMO), Tonkolili district, shared findings from the pillar level reviews to a wider audience. The lead facilitator guided plenary discussions that followed each presentation and the audience provided inputs on additional issues and other aspects that may have contributed to successes or gaps in the outbreak response. Ultimately, the discussions from the bigger group served to improve, adopt or revise the findings and recommendations from pillar level AARs in order to ensure that there was objectivity and impartiality in the final outcome.

All the inputs during the plenary session were compiled by note takers who had been selected for this session. The notes were then handed over to the lead facilitator who compiled all the AAR findings into a comprehensive report that was disseminated to all stakeholders. All pillars were then requested to meet later to develop action plans based on the final recommendations made during the AAR.

## Results

### Coordination of the response at national and district level

Sierra Leone has an established Public Health National Emergency Operation Center (PHNEOC) in the capital city Freetown and District Emergency Operation Centres (DEOC) in all 16 districts. Immediately following the notification of confirmed Lassa fever on November 20, 2019, the MOHS activated PHNEOC and a multiagency incident management system

**Table 2. Guiding questions for after action review of Lassa fever outbreak, Tonkolili district, Sierra Leone, 2019.**

a) What was expected to happen
b) What actually Happened
c) What worked well
d) What didn't work well
e) Why was there a difference?
f) What can be learned

was initiated to coordinate national and district level response activities, resource management, communication, and information management.

The incident management system was organized into six main pillars namely 1) Coordination; 2) Surveillance and laboratory; 3) Case management; 4) IPC, safe and dignified burial; 5) Risk communication and social mobilization; and 6) Logistics. An incident manager was nominated to provide strategic leadership and direction for the pillars. The pillar leads were responsible for leading and coordinating national and district-level activities across each of the six pillars in the incident management system.

During the AAR, coordination of the response was found to be well organized at the national and district level. The district Rapid Response Team (RRT) was dispatched to the field within 24 hours of Lassa Fever notification as required. The national Rapid Response Team (RRTs) arrived after 48 hours to support the district team although it did not include a veterinarian and as such did not comply with One Health approach.

The MoHS mobilized resources for the outbreak response in time, although initial funding from government was delayed and hence partners provided the initial support. District RRT generated daily situational reports on the outbreak response and this supported informed decision making at both DEOC and PHNEOC. Notable gaps in coordination included deployment of the National Incident Manager to the district (as part of the national RRT), resulting in a significant gap in strategic oversight at the national level. Poor internet connectivity in Tonkolili district also affected timely communications with the national level.

## Case management and IPC

Having identified the potential breach in IPC guidelines in the affected hospital, National IPC unit staff with support from WHO country office staff identified core activities to mitigate further spread of the outbreak. Cleaning and disinfection of Masanga Hospital, re-training of health workers on IPC and triaging of patients and provision of IPC supplies were some of the initial response activities. Refresher trainings were also conducted among health workers in surrounding health facilities and included re-sensitization on recommended standard precautions for health care settings.

Notable gaps included lack of IPC supplies such as soap and water in some health facilities and low level of knowledge on how to use some of the IPC and personal protective equipment (PPE) supplies. In contrast, health workers in Masanga Hospital were reported not to always use protective gear despite the hospital having enough stocks of PPE. Incorrect decontamination of medical devices, poor environmental cleanliness, non-compliance to hand hygiene and inadequate waste management were other gaps observed. These findings contrasted reports of high IPC compliance in Masanga Hospital based on prior audits used to assess compliance to a minimum set of IPC standards. Neither Tonkolili District Health Management Team nor the hospitals included in the assessments had fully functional IPC Committees.

Lack of a technical lead to guide the management of cases at the district level was a concern both during the outbreak response and the after-action review. However, cases were managed in a neighboring district at the Kenema Government Hospital Lassa Fever unit which is well equipped to manage Lassa Fever cases. The only surviving confirmed case in the country was treated in this unit using intravenous Ribavirin and recovered fully.

## Surveillance and laboratory

Whereas the surveillance response to the outbreak was sufficient in identifying additional cases and tracking contacts, it was observed that there was delay in detecting the primary cases in Masanga hospital. This is because Lassa Fever was not regularly considered as a differential

diagnosis among patients with acute febrile illness in the district despite the disease being endemic in some parts of the country. However, Acute Viral Hemorrhagic Fever (AVHF) is a notifiable disease in Sierra Leone and all patients with fever and bleeding are supposed to be tested for AVHF which includes Lassa Fever, Ebola, Marburg etc. Therefore, if this case was picked as an AVHF then detection would have been faster.

Once confirmation of the first case of Lassa Fever was made, the surveillance teams undertook active case search in health facilities and communities, and identified 4 more cases. A total of 81 contacts were also identified in Sierra Leone and were then successfully followed up for 21 days. Integrated disease surveillance and response guidelines and rapid response guidelines became important references that allowed a structured approach to the response. The team liaised with the community mobilization teams, local leaders and security officers to counter initial community resistance to contact tracing. Six laboratory samples were collected and tested at the Kenema Government Hospital laboratory in neighboring Kenema district and turnaround time was less than 48 hours for all samples. Inadequate supply of sample collection materials delayed sample collection initially, but this was eventually resolved. At the time of this outbreak, only Kenema Government Hospital laboratory was able to test for Lassa Fever as the two other reference laboratories in the country were unable to run PCR tests for Lassa Fever due to shortage of reagents.

### Risk communication and social mobilization

Risk communication and health education messages were incorporated into the response activities from the onset of the response and continued throughout, addressing misconceptions and fears in the community. Information booths were set up in the two affected communities and radio dialogues and audios in local languages were aired. This intervention was considered critical and assuaging to the initial reaction from the community due to misconceptions related to the past Ebola outbreak management in that locality. Press briefings at the national and district level also provided updates to the public. Media monitoring on mainstream media and social media was conducted and rumor countered appropriately. Despite this mostly efficient communication, sometimes the communications team did not receive information from the technical teams on time and this in turn delayed the onward communication of information to the public.

### Logistics pillar

Although response supplies (including for risk communication, IPC and laboratory specimen collection) were provided to the district, it took more than 48 hours to deliver them from the national level after the notification of the outbreak was made. This was partly due to poor information management between district and PHNEOC logistics team, lack of pre-quantification of response needs and lack of contingency funding. Partner coordination at the district level was well done although it remained unclear what resources were made available by district level partners.

Table 3 provides a summary of the best practices and lessons learnt by pillar. Based on the lessons learnt and challenges encountered during the response, a total of 60 recommended actions were made during the AAR which were distributed as follows: Coordination 8; Case management and IPC 8; Surveillance 9; Laboratory 4; Risk communication and social mobilization 8; Logistics 7; District Health Management Team 16. Each pillar was requested to develop detailed implementation plans based on these recommended actions.

### Discussion

According to World Health Organization guidelines, An AAR should be carried out within 3 months of the end of a public health event [1]. The AAR of the response to the Lassa Fever

**Table 3. Best practices and lessons learnt during Lassa fever outbreak response, Tonkolili district, Sierra Leone, 2019.**

| Action | Best practices | Lessons learnt |
|---|---|---|
| Coordination of outbreak Response (National and District Level) | ● District rapid response team deployed within 24 hours of Lassa Fever notification<br>● Regular coordination meetings held at national and district emergency operation centres<br>● Daily situational reports issued by the district<br>● Timely brief by senior ministry leadership provided credible information to the public, allaying anxiety | ● The national and district rapid response teams are an invaluable asset on stand-by<br>● Good collaboration between district and national level teams is required during emergencies<br>● Timely access to government resources can be a challenge during emergencies and needs to be addressed |
| Surveillance and Contact Tracing | ● Active case search was conducted in four health facilities in Tonkolili district and affected communities.<br>● All contacts identified were successfully followed up for 21 days<br>● Collaboration with local leaders overcame resistance to contact tracing | ● Low index of suspicion among clinical staff can delay case detection of Lassa fever<br>● Lassa Fever should be included as a differential diagnosis for all pregnant women who die with fever due to the high maternal mortality in Sierra Leone. A blood sample should therefore be taken for testing before burial |
| Case Management | ● Kenema Government Hospital Lassa Fever Unit was promptly identified and used for isolation and management of all Lassa Fever suspected cases | ● A dedicated Lassa Fever Treatment unit is an invaluable asset for the country<br>● Late diagnosis of Lassa fever may be the underlying reason for the unusually high CFR in Sierra Leone |
| Infection Prevention and Control | ● Assessment of IPC compliance and availability of IPC supplies in selected health facilities in Tonkolili District<br>● Refresher training of 70 health care workers on IPC conducted in Tonkolili district<br>● Distribution of IPC supplies in peripheral health units | ● Low IPC compliance among staff can cause costly hospital associated outbreaks. Compliance should be enforced through regular audits<br>● IPC compliance audits should also focus on IPC practice in addition to health workers' knowledge |
| Laboratory | ● Well defined sample collection protocols and sample referral networks ensured rapid turnaround time | ● Diagnosis of Lassa Fever is a challenge and good investment in point of care diagnostics is required |
| Risk Communication and Social Mobilization | ● Pre-designed risk communication messages quickly customized for the situation<br>● Use jingles (radio messages) in local languages ensured public health messages were ideal for the affected communities<br>● Media monitoring identified counter-productive messages circulating on social media and countered them in a timely manner | ● Use of local leaders is important in overcoming resistance from communities during response |
| Logistics and Operations | ● Ministry of Health and Sanitation successfully applied for funds from government to respond to outbreak although received late | ● Good coordination is required during planning and delivery of logistics from national to district |

outbreak in Sierra Leone was conducted within three months after the end of the outbreak and provided insight on the status of preparedness and response in the country. This was particularly important as Lassa fever was the highest ranking health risk in Sierra Leone. At the time of the review, Nigeria, where Lassa Fever is endemic, was also experiencing a widespread outbreak [16]. Therefore, there were concerns that Sierra Leone, like other countries in the Mano River Union prone to Lassa Fever outbreaks, could also experience a surge in cases.

This outbreak likely resulted from a breach in IPC protocols due to the fact that all four secondary cases were related to a single surgical operation. Infections and deaths among health workers have occurred occasionally in other countries such as Nigeria when adherence to barrier nursing and contact precautions are not maintained [17]. One such outbreak occurred in Nigeria in a health facility in 2018 where sixteen health workers were infected with a case fatality rate (CFR) of 31.6%. This fatality rate is quite similar to the Sierra Leone incident where the total CFR among the five cases was 60% but only one of three health workers infected died (CFR 33%). The total number of laboratory confirmed Lassa Fever cases reported in Sierra Leone was 15 to 35 cases annually from 2016 to 2019. The positivity rate of suspected cases ranged between 6% and 12% in those 4 years while overall CFR was 39% but higher among admitted patients at 63% in the four years. We believe that conducting regular AARs should help unearth challenges which if addressed can help reduce the high fatality rate.

During the AAR, areas that were found to have worked well in the outbreak response included coordination at the national and the district level. This is an important improvement as poor district level coordination structures were a concern during the mudslide emergency response in 2017 [11]. Response from health sector partners was timely and well-coordinated by MoHS national level team. Once the outbreak was confirmed, technical pillars, including surveillance, laboratory, IPC, Risk Communication and Social Mobilization launched an effective response that rapidly contained further spread of the outbreak.

Despite previous assessment reports showing favorable status of IPC in the affected hospital, the AAR established gaps in use of Personal Protective Equipment (PPEs) by staff and low level of knowledge on IPC among health care workers. This raises concerns on the effectiveness of routine assessments to identify gaps in IPC, especially as the assessments are more quantitative, focus on knowledge measurement and gauging availability of IPC commodities and do not include observation of IPC practices at the ward level.

Lack of dedicated district level and hospital level IPC committees was noted to reduce visibility and focus on IPC. An assessment of IPC status conducted in late 2014, during the Ebola outbreak, also identified lack of IPC focal persons at the hospital and district level as a challenge [18]. Although IPC focal persons are now available in most public health facilities, a dedicated committee would add the necessary impetus to IPC matters and increase chances of change in IPC compliance. Despite these challenges in IPC, rapid institution of control measures limited the further spread of the outbreak. Increasing compliance of health facility staff to standard precautions at all times is critical, given that highly infectious and life-threatening pathogens have been shown to circulate unidentified in Sierra Leone [19].

Availability and reference to pre-existing preparedness and response structures including an incident management system allowed for prompt response to the outbreak. However, deployment of the national incident manager as part of the RRT left a significant gap in strategic oversight and management of the incident at the national level. Resource mobilization was a challenge with government funding coming in late in the response. Although partners such as WHO stepped in to fill the funding gap, this raises concerns over sustainability during future emergencies. Whereas government funding for emergencies in Sierra Leone has increased over time, rapid availability of funds during emergencies remains challenging, partly due to the financial accountability and bureaucratic procedures in place. Funding constraints are reported as a major challenge affecting emergency response in low resource settings [20] while in contrast, higher investments in public health emergency preparedness and response appears to be paying off in China [21].

Among the prominent shortcomings identified during the review was delayed case detection. Several factors likely contributed to this including a low index of suspicion among attending clinicians, lack of readily available rapid diagnostic testing platforms and failure to conduct records review to identify suspected Lassa Fever cases. At the time of the outbreak, diagnosis of Lassa Fever in Sierra Leone was conducted mainly at the regional Laboratory in Kenema Government Hospital in the east of the country or the Central Public Health Reference Laboratory in the capital city Freetown. Laboratory specimens collected in Masanga Hospital would likely be referred to the laboratory in Kenema due to proximity. However, the clinician would first have to suspect Lassa Fever, partly a diagnosis of exclusion in the early stages of the illness, before they make the decision to collect a specimen for laboratory confirmation. If there are challenges in specimen referral, such as lack of transport then this can influence the decision to collect specimens.

The AAR identified challenges in specimen collection and transportation during the response and this supports the proposition that routine testing for Lassa Fever can be challenging in this district. Severe Lassa Fever in pregnancy is a difficult diagnosis as it is not always

high on the list of probable causes of bleeding. Clinicians are likely to first consider more common causes of vaginal bleeding in pregnancy. Although failure to review medical records for priority conditions, including Lassa fever was a shortcoming, this exercise is only useful if clinicians use standard case definitions to record cases in the medical records. The occurrence of fever alongside bleeding should always be regarded by the clinician with a high index of suspicion.

Whereas the value of AAR is widely acclaimed, some factors hinder regular reviews of the response to public health events and the usefulness of AAR in bringing meaningful change is therefore not always realized. These include fear of being reprimanded, caution about exposing sensitive health security matters, constraints in bringing stakeholders together in a blame free environment and time and resource constraints. Additionally, failure to implement recommendations from AARs may result in repetition of the same mistakes over and over again.

To overcome some of these factors, the preliminary pillar level reviews during this AAR allowed more open and intense discussions in smaller groups. This was intended to allow the participants to express themselves more freely. Moreover, assessments of preparedness and response functionality are common in Sierra Leone with regular simulation exercises, and annual in-depth assessment of IHR capacities though the annual state party IHR reporting and self-assessment using the Joint External Evaluation score-card.

This paper has some limitations. First, some of the information collected was based on self-reports that could have been biased especially among the hospital staff who may have been expected to take "best practice" actions but did not. This bias was reduced by triangulating information from other staff and other reports as much as possible. Second, although recommendations were made for each pillar during the AAR, detailed action plans for implementation were not completed during the meeting. Information was therefore not available on how many of the pillars had developed the action plans and how many of the 60 recommendations had been implemented by the time of writing this article. It is possible that some of the recommendations had been implemented but no documentation was available.

## Conclusion

This After Action Review article provides best practices and lessons learnt in response to a Lassa Fever outbreak in Sierra Leone and provides insight on the functionality of the system to deal with outbreaks of epidemic prone diseases. Coordination of the outbreak response at the national and the district level was relatively well done. Conversely, delayed case detection was a concern and has implications for the sensitivity of the surveillance system to detect other high priority conditions. Clinicians therefore require regular training on surveillance case definitions so as to increase case detection for Lassa Fever as well as other priority conditions. This should be complemented by a functional sample referral system.

The national and subnational IPC programs should review current assessment methods and implement regular audits of IPC practice that yield representative findings. Additionally, mechanisms should be put in place to ensure timely disbursement of funding for emergency response while striking a balance on the need for financial accountability. Lastly, implementation of recommendations from the AAR should be tracked and documented for future reference.

## Supporting information

**S1 File. Learning from Incidents and Exercises: Guidance on EPRR Debriefing and After Action Reviews.**
(PDF)

**S2 File. Reporting Template for After Action Review.**
(PDF)

## Acknowledgments

We acknowledge the contributions of the staff from the Sierra Leone Ministry of Health and Sanitation, Tonkolili District Health Management Team, World Health Organization, US Centers for Disease Control and Prevention and all other organizations that participated in the After Action Review meetings that generated data for this article.

## Author Contributions

**Conceptualization:** Charles Njuguna, Mohamed Vandi, Evans Liyosi, Jane Githuku, James Sylvester Squire, Ian Njeru, Ian Rufus, Victoria Katawera, Wilson Gachari, Robert Musoke, Claudette Amuzu, Mukeh Fahnbulleh, Joseph Bunting-Graden, Janet Kayita, James Bunn, Ambrose Talisuna, Zabulon Yoti.

**Data curation:** Charles Njuguna, Jane Githuku, James Sylvester Squire, Ian Njeru.

**Formal analysis:** Jane Githuku, Ian Njeru.

**Funding acquisition:** Charles Njuguna.

**Investigation:** Charles Njuguna, Mohamed Vandi, Jane Githuku, James Sylvester Squire, Ian Njeru, Ian Rufus, Victoria Katawera, Wilson Gachari, Robert Musoke, Claudette Amuzu, Mukeh Fahnbulleh, Joseph Bunting-Graden, Janet Kayita, James Bunn.

**Methodology:** Charles Njuguna, Jane Githuku, Ian Njeru, Ian Rufus, Robert Musoke, Claudette Amuzu, Mukeh Fahnbulleh, Joseph Bunting-Graden.

**Project administration:** Charles Njuguna, Robert Musoke.

**Resources:** Charles Njuguna.

**Software:** Jane Githuku, Ian Njeru.

**Supervision:** Charles Njuguna, Mohamed Vandi, Evans Liyosi, Robert Musoke.

**Validation:** Ian Rufus.

**Visualization:** Jane Githuku, Ian Njeru.

**Writing – original draft:** Jane Githuku.

**Writing – review & editing:** Charles Njuguna, Mohamed Vandi, Evans Liyosi, Jane Githuku, James Sylvester Squire, Ian Njeru, Ian Rufus, Victoria Katawera, Wilson Gachari, Robert Musoke, Claudette Amuzu, Mukeh Fahnbulleh, Joseph Bunting-Graden, Janet Kayita, James Bunn, Ambrose Talisuna, Zabulon Yoti.

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
