## [Decision Letter · Decision Letter 0]

22 Apr 2022

Dear Dr Njuguna,

Thank you very much for submitting your manuscript "After action review of the response to an outbreak of Lassa fever in Sierra Leone, 2019: Best Practices and Lessons Learnt" for consideration at PLOS Neglected Tropical Diseases. As with all papers reviewed by the journal, your manuscript was reviewed by members of the editorial board and by several independent reviewers. In light of the reviews (below this email), we would like to invite the resubmission of a significantly-revised version that takes into account the reviewers' comments. 

In addition, please consider the following:

1) Adequate justification is needed why the lead AAR team members were part of the Ministry of Health, which was leading implementation. The impact of this on the reported findings and measures taken to mitigate this need to be adequately justified in the manuscript.

2) Action taken after the AAR was not clear. AAR is not evaluation, so the follow up action is very important. It appears from the paper that the authors have developed an Action Plan at the end, but the content of the stated action plan is unclear.

We cannot make any decision about publication until we have seen the revised manuscript and your response to the reviewers' comments. Your revised manuscript is also likely to be sent to reviewers for further evaluation.

Sincerely,

Fasil Tekola-Ayele

Associate Editor

Andrea Marzi

Deputy Editor

Reviewer's Responses to Questions

**Key Review Criteria Required for Acceptance?**

**Methods**

-Are the objectives of the study clearly articulated with a clear testable hypothesis stated?

-Is the study design appropriate to address the stated objectives?

-Is the population clearly described and appropriate for the hypothesis being tested?

-Is the sample size sufficient to ensure adequate power to address the hypothesis being tested?

-Were correct statistical analysis used to support conclusions?

-Are there concerns about ethical or regulatory requirements being met?

Reviewer #1: Although this article does not follow the classical structure of a Research paper (and how should it?), there is a very clear structure and the important aspects of the results are picked up in the discussion. Altogether, the Methods section could be shortened a bit to make it more readable.

As the objective was to cover the AAR, the design of the manuscript is adequate

It would be informative to the Reader to have the template used for reporting available as supplementary material. A function of the paper would also be to guide future AARs with respect to the application in different Settings.

No ethical concerns.

Reviewer #2: (No Response)

**Results**

-Does the analysis presented match the analysis plan?

-Are the results clearly and completely presented?

-Are the figures (Tables, Images) of sufficient quality for clarity?

Reviewer #1: p. 12 first Paragraph. the authors found that it was failed to suspect LF as a differential in the index case with Fever and vaginal bleeding. While it is absolutely important to state that this presentation is at high Risk for LF, it is also true that HCW can fail to define the suspicion for a multitude of reasons. It would be important to know if there are any Guidelines or if there is any case Definition (for non-outbreak Situation probably) that could have helped make this differential diagnosis or anything that could guide early case detection in SL. it is stated in Table 3 that it should be included as a differential diagnosis in pregnant women who die, however it would be important to define how this Information can be delivered to HCW in a better Fashion.

P.14 l. 293. Which form of Ribavirin was used? p.o. / i.v.

Reviewer #2: (No Response)

**Conclusions**

-Are the conclusions supported by the data presented?

-Are the limitations of analysis clearly described?

-Do the authors discuss how these data can be helpful to advance our understanding of the topic under study?

-Is public health relevance addressed?

Reviewer #1: Conclusions are very clearly presented. Limitations not so clearly. What could have been achieved more in the AAR?

Public Health relevance is clearly adressed.

Reviewer #2: Some of the conclusions not supported by the data provided. Limitations of the study have not been included

**Editorial and Data Presentation Modifications?**

Reviewer #1: The manuscript is written very clearly and concisely, and is by itself informative. It would be very helpful if the authors could share some of the Underlying documents 7 templates / Guidelines as suggested above and below. 

Altogether, i suggest minor revisions

Reviewer #2: (No Response)

**Summary and General Comments**

Reviewer #1: minor comments:

- p. 10 last Paragraph. It would be interesting to Quote the source of the Country Guidelines or even offer them as supplementary material it this is allowed, to inform the Reader of the Framework of the AAR The same is true for p. 11 line 217 concerning the templates

the mortality is very high in this small cluster, compared to the mortality in endemic Areas. Maybe the authors could compare this to data published for endemic / non-endemic Areas? are there reasons to believe that mortality could be improved due to implementation of AAR results?

Reviewer #2: The manuscript is based on an after-action review (AAR) of a Lassa fever outbreak in Sierra Leone as a mechanism for improving on preparedness and response actions to public health outbreaks. The AAR is an important evaluation of disease outbreak response and documenting such findings could be useful in countries with similar contexts.

Major comment

My main concern is that the manuscript does not follow appropriately or provide the information expected in each section of a research article. It seems to be a written as a report which was then loosely attempted to be turned into a manuscript. 

Other comments

Abstract: The authors state that the outbreak was caused by a breach of IPC protocols. No evidence is provided in the results to back this assertion.

Line 66: It is not apparent to the reader that the simulation exercise and capacity assessments the authors refer to are among the four components of the IHR monitoring framework

Line 68: Are AARs part of organizational learning? That needs to be clarified so that the paragraph can have coherence

Line 76: Use the appropriate citation style for the journal 

Line 92: Give a reference for the sentence 

Line 90 to 104: The two paragraphs are redundant and repetitive. The rationale for AARs is already stated in the introductory paragraph.

Line 116: Incident not incidence

Line 121: Wasn’t the outbreak occurrence already known after the notification from Netherlands? 

Line 129 to 143: This description of the response should be in the results section.

Line 171: This subsection should be made brief and moved to before the incident description

Line 183: The table should be moved to methods

Line 190-193: Did the authors use the WHO guidance on AAR as well?

Line 196: What were the considerations or qualities of the lead facilitator? Experience in AAR?

Line 204: What did the presentation template for the pillar discussions contain? Alternatively include it as a supplementary document.

Line 206: Give a reference for the document. Were the participants trained on conducting AAR besides being provided with the document.

Line 229: The coordination description is already stated in the introduction and methods should be provided before concluding with this sentence.

Line 332: Table 3 section on surveillance and contact tracing -- Lassa Fever should be included

as a differential diagnosis for all pregnant women who die. 

 What is the rationale for including ALL pregnant women? Including those dying from say eclampsia? With the limited Lassa fever testing, how would this be implemented?

Line 351: The data presented does not give evidence to the strong assertion on the outbreak attribution.

Discussion:

Include limitations in your current study that could affect interpretation of the findings

PLOS authors have the option to publish the peer review history of their article (what does this mean?). If published, this will include your full peer review and any attached files.

Reviewer #1: No

Reviewer #2: No
---

## [Decision Letter · Decision Letter 1]

11 Jul 2022

Dear Dr Njuguna,

Thank you very much for submitting your revised manuscript "After action review of the response to an outbreak of Lassa fever in Sierra Leone, 2019: Best Practices and Lessons Learnt" for consideration at PLOS Neglected Tropical Diseases. As with all papers reviewed by the journal, your manuscript was reviewed by members of the editorial board and by several independent reviewers. The reviewers appreciated the attention to an important topic. Based on the reviews, we are likely to accept this manuscript for publication, providing that you modify the manuscript according to the reviewers' recommendations. 

Sincerely,

Fasil Tekola-Ayele

Academic Editor

Andrea Marzi

Section Editor

Reviewer's Responses to Questions

**Key Review Criteria Required for Acceptance?**

**Methods**

-Are the objectives of the study clearly articulated with a clear testable hypothesis stated?

-Is the study design appropriate to address the stated objectives?

-Is the population clearly described and appropriate for the hypothesis being tested?

-Is the sample size sufficient to ensure adequate power to address the hypothesis being tested?

-Were correct statistical analysis used to support conclusions?

-Are there concerns about ethical or regulatory requirements being met?

Reviewer #2: Adequate for the manuscript

**Results**

-Does the analysis presented match the analysis plan?

-Are the results clearly and completely presented?

-Are the figures (Tables, Images) of sufficient quality for clarity?

Reviewer #2: Adequate for the manuscript

**Conclusions**

-Are the conclusions supported by the data presented?

-Are the limitations of analysis clearly described?

-Do the authors discuss how these data can be helpful to advance our understanding of the topic under study?

-Is public health relevance addressed?

Reviewer #2: Adequate

**Editorial and Data Presentation Modifications?**

Reviewer #2: (No Response)

**Summary and General Comments**

Reviewer #2: The manuscript has improved from the previous versions and the reviewer comments have mostly been addressed. 

See below minor comments. In lieu of absent line numbers, i have tried to be as descriptive as possible on the referenced lines or sentences. 

Abstract page 2, results, first sentence: Causation on an IPC breach resulting in transmission of Lassa fever was not established based on the presented results. Use terms such as associated, likely occurred. This comment applies to the other sections of the manuscript where similar assertions are made. It is important not to use language that imply causation without the requisite data or evidence.

Introduction section, page 4, first paragraph: The manuscript is lengthy and removing this paragraph would help because the paragraph is unnecessary because the aim of AAR is already well stated in the previous paragraph

Introduction section, page 6, 3rd last line: Did this patient have a febrile illness at the time of admission or surgery? This will help the reader appreciate if the patient could have already been infected.

Methods, page 7 second last line: The MoHS guidance included in the supplementary provides that the AARs are within 6 weeks of the incident. Clarify why it is inferred in the current text that AARs are expected to be conducted within three months. Is it in reference to a guidance different from that from MoHS.

Page 12, case management and IPC section, first line: Was there specific information of IPC breaches during the surgery on the index case? Is it that the surgical team did not use gloves, surgical gowns or masks? 

Page 16, Discussion first two sentence: Remove sentence section after the comma because it is redundant.

Page 17, discussion, first paragraph, line 8 and 9: rewrite the cases by year for ease of readability. .. “15 to 35 cases annually between 2016-2019” 

Page 20, second last paragraph on limitations: An important limitation to include in this study is that the some of the data was based on self-reports that could be biased especially among the staff who may have been expected to take “best practice” actions but did not. There might have been fear of "self-incrimination". The bias could have been reduced by triangulating the information with reports and documentation during the outbreak and before the AAR.

PLOS authors have the option to publish the peer review history of their article (what does this mean?). If published, this will include your full peer review and any attached files.

Reviewer #2: No

Figure Files:

Data Requirements:

Reproducibility:

References

---

## [Decision Letter · Decision Letter 2]

19 Aug 2022

Dear Dr Njuguna,

We are pleased to inform you that your manuscript 'After action review of the response to an outbreak of Lassa fever in Sierra Leone, 2019: Best Practices and Lessons Learnt' has been provisionally accepted for publication in PLOS Neglected Tropical Diseases.

Best regards,

Andrea Marzi

Section Editor

Reviewer's Responses to Questions

**Key Review Criteria Required for Acceptance?**

**Methods**

-Are the objectives of the study clearly articulated with a clear testable hypothesis stated?

-Is the study design appropriate to address the stated objectives?

-Is the population clearly described and appropriate for the hypothesis being tested?

-Is the sample size sufficient to ensure adequate power to address the hypothesis being tested?

-Were correct statistical analysis used to support conclusions?

-Are there concerns about ethical or regulatory requirements being met?

Reviewer #2: Adequate

**Results**

-Does the analysis presented match the analysis plan?

-Are the results clearly and completely presented?

-Are the figures (Tables, Images) of sufficient quality for clarity?

Reviewer #2: Adequate

**Conclusions**

-Are the conclusions supported by the data presented?

-Are the limitations of analysis clearly described?

-Do the authors discuss how these data can be helpful to advance our understanding of the topic under study?

-Is public health relevance addressed?

Reviewer #2: Adequate

**Editorial and Data Presentation Modifications?**

Reviewer #2: Accept

**Summary and General Comments**

Reviewer #2: The auhtors have addressed the comments raised from the previous revision.

PLOS authors have the option to publish the peer review history of their article (what does this mean?). If published, this will include your full peer review and any attached files.

Reviewer #2: No

---

## [Editor Report · Acceptance letter]

9 Sep 2022

Dear Dr Njuguna,

We are delighted to inform you that your manuscript, "After action review of the response to an outbreak of Lassa fever in Sierra Leone, 2019: Best Practices and Lessons Learnt," has been formally accepted for publication in PLOS Neglected Tropical Diseases.

Best regards,

Shaden Kamhawi

co-Editor-in-Chief

Paul Brindley

co-Editor-in-Chief
